# Non-Stationary Contextual Bandit Learning via Neural Predictive Ensemble Sampling

## Abstract

Real-world applications of contextual bandits often exhibit non-stationarity due to seasonality, serendipity, and evolving social trends. While a number of non-stationary contextual bandit learning algorithms have been proposed in the literature, they excessively explore due to a lack of prioritization for information of enduring value, or are designed in ways that do not scale in modern applications with high-dimensional user-specific features and large action set, or both. In this paper, we introduce a novel non-stationary contextual bandit algorithm that addresses these concerns. It combines a scalable, deep-neural-network-based architecture with a carefully designed exploration mechanism that strategically prioritizes collecting information with the most lasting value in a non-stationary environment. Through empirical evaluations on two real-world recommendation datasets, which exhibit pronounced non-stationarity, we demonstrate that our approach significantly outperforms the state-of-the-art baselines.

## 1 Introduction

Contextual bandit learning algorithms have seen rapid adoptions in recent years in a numder of domains (Bouneffouf and Rish, 2019), from driving personalized recommendations (Li et al., 2010) to optimizing dyanmic advertising placements (Schwartz et al., 2017). The primary objective of these algorithms is to strategically select actions to acquire information about the environment in the most cost-effective manner, and use that knowledge to guide subsequent decision-making. Thanks in part to the historical development in this field, many of these algorithms are designed for a finite-horizon experiment with the environment remaining relatively stationary throughout.

However, real-world environments are rife with non-stationarity (Ditzler et al., 2015; Elena et al., 2021), as a result of seasonality (Keerthika and Saravanan, 2020; Hwangbo et al., 2018), serendipity (Kotkov et al., 2016; 2018), or evolving social trends (Abdollahpouri et al., 2019; Cañamares and Castells, 2018). To make matters worse, many practical contextual bandit systems, such as these commonly used in a recommendation engine, operate in a continuous manner over a long, or even indefinite time horizon, further exposing the learning algorithm to non-stationarity that is bound to manifest over its lifetime. Indeed, when applied to non-stationary environments, traditional contextual bandit learning algorithms designed with stationarity in mind are known to yield sub-optimal performance (Trovo et al., 2020; Russac et al., 2020).

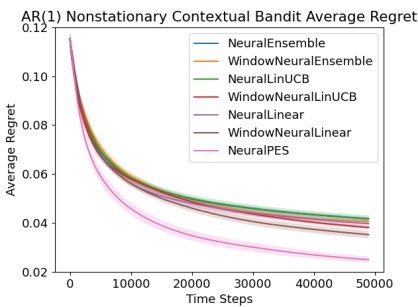

Figure 1: NeuralPES Regret in Nonstationary Contextual Bandits

The goal of this paper is to study the design of contextual bandit algorithms that not only successfully navigate a non-stationary environment, but also scale to real-world production environments. Extending classic bandit algorithms to a non-stationary setting has received sustained attention in recent years (Kocsis and Szepesvári, 2006; Garivier and Moulines, 2008; Raj and Kalyani, 2017; Trovo et al., 2020). A limitation in these existing approaches, however, is that they are designed to explore new information by only decaying past information, which could introduce excessive exploration. As pointed out by Liu et al. (2023), exploration designs intended for stationary environments tend to focus on resolving the uncertainty surrounding an action's current quality, and as

such, suffer sub-optimal performance for failing to prioritize collecting information that would be of more enduring value in a non-stationary environment. In response, Liu et al. (2023) proposed the predictive sampling algorithm that takes information durability into account, and demonstrated an impressive performance improvement over existing solutions. However, the predictive sampling algorithm, among many nonstationary contextual bandit learning algorithm we discuss in the related work section, suffers from their scalability and does not scale with modern deep learning systems.

In this work, we take a step towards solving large-scale nonstationary contextual bandit problems by introducing Neural Predictive Ensemble Sampling (NeuralPES), the first non-stationary contextual bandit learning algorithm that is scalable with modern neural networks and effectively explores in a non-stationary envrionment by seeking lasting information. Theoretically, we establish that NeuralPES emphasizes the acquisition of lasting information, information that remains relevant for a longer period of time. Empirically, we validate the algorithm's efficacy in two real-world recommendation datasets, spanning across 1 week and 2 months of time, respectively, and exhibiting pronounced non-stationarity. Our findings reveal that our algorithm surpasses other state-of-the-art neural contextual bandit learning algorithms, encompassing both stationary and non-stationary variants. As a spoiler for our empirically results, see Figure 1 for the average regret of our agent compared to other baselines on an AR(1) nonstationary contextual bandit environment.

## 2 RELATED WORK

**Non-Stationary Bandit Learning.** A large number of non-stationary bandit learning algorithms rely on heuristic approaches to reduce the effect of past data. These heuristics include maintaining a sliding window Cheung et al. (2019; 2022); Garivier and Moulines (2008); Russac et al. (2020); Srivastava et al. (2014); Trovo et al. (2020), directly discounting the weight of past rewards by recency Bogunovic et al. (2016); Garivier and Moulines (2008); Russac et al. (2020); Kocsis and Szepesvári (2006), restarting the algorithm periodically or with a fixed probability at each time Auer et al. (2019a); Allesiardo et al. (2017); Besbes et al. (2019); Bogunovic et al. (2016); Wei et al. (2016); Zhao et al. (2020), restarting upon detecting a change point Abbasi-Yadkori et al. (2022); Allesiardo and Féraud (2015); Auer et al. (2019b); Allesiardo et al. (2017); Besson and Kaufmann (2019); Cao et al. (2019); Chen et al. (2019); Ghatak (2021); Ghatak et al. (2021); Hartland et al. (2006); Liu et al. (2018); Luo et al. (2018); Mellor and Shapiro (2013), and more complex heuristics (Gupta et al., 2011; Kim and Tewari, 2020; Raj and Kalyani, 2017; Viappiani, 2013). These algorithms adapt stationary bandit learning algorithms like Thompson sampling (TS) (Thompson, 1933), Upper Confidence Bound (UCB) (Lai and Robbins, 1985), and exponential-weight algorithms (Rexp3) (Auer et al., 2002; Freund and Schapire, 1997) using aforementioned heuristics to reduce the impact of past data and encourage continual exploration. However, they often lack intelligent mechanisms for seeking lasting information during exploration. While predictive sampling (Liu et al., 2023) seeks for lasting information, it does not efficiently scale.

**Deep Neural Network-Based Bandit Algorithms.** In practical applications of bandit learning, both the set of contexts and the set of actions can be large. A number of algorithms (Gu et al., 2021; Jia et al., 2022; Kassraie and Krause, 2022; Riquelme et al., 2018; Salgia, 2023; Su et al., 2023; Xu et al., 2022; Zhang et al., 2020; Zhou et al., 2020; Zhu and Van Roy, 2023b) utilize the capacity of deep neural networks to generalize across actions and contexts. These algorithms are designed for stationary environments. While Allesiardo et al. (2014) proposes a deep neural-network based algorithm for non-stationary environments, it does not intelligently seek for lasting information.

## 3 CONTEXTUAL BANDITS

This section formally introduces contextual bandits, and other related concepts and definitions. We first introduce contextual bandits.

**Definition 1** (**Contextual Bandit**). A contextual bandit $\mathcal{E}$ with a finite set of contexts $\mathcal{C}$ and a finite set of actions $\mathcal{A}$ is characterized by three stochastic processes: the reward process $\{R_t\}_{t \in \mathbb{N}}$ with state space $\mathbb{R}^{|\mathcal{C}|} \times \mathbb{R}^{|\mathcal{A}|}$, the contexts $\{C_t\}_{t \in \mathbb{N}}$ with state space $\mathcal{C}$, and the sequence of available action sets $\{\mathcal{A}_t\}_{t \in \mathbb{N}}$ with state space $2^{\mathcal{A}}$. We use $\mathcal{E} = (\{R_t\}_{t \in \mathbb{N}}, \{C_t\}_{t \in \mathbb{N}}, \{\mathcal{A}_t\}_{t \in \mathbb{N}})$ to denote the bandit.

At each timestep $t \in \mathbb{N}$, an agent is presented with context $C_t$ and the set of available actions $\mathcal{A}_t$. Upon selecting action $a \in \mathcal{A}_t$, the agent observes a reward of $R_{t+1,C_t,a}$.

### 3.1 LINEAR CONTEXTUAL BANDITS

In many practical applications, both the context set and the action set are large. To enable effective generalization across these sets, certain structural assumptions on how the rewards are generated

come into play. In this regard, the reward $R_{t,c,a}$ can be described as a function of a feature vector $\phi(c, a)$, which captures relevant contextual information in context $c \in \mathcal{C}$ and action information in action $a \in \mathcal{A}$. To exemplify this structure, let us introduce the linear contextual bandit.

**Example 1** (**Linear Contextual Bandit**). A linear contextual bandit is a contextual bandit with feature mapping $\phi : \mathcal{C} \times \mathcal{A} \to \mathbb{R}^d$, a stochastic process $\{\theta_t\}_{t \in \mathbb{N}}$ with state space $\mathbb{R}^d$. For all $t \in \mathbb{N}$, $c \in \mathcal{C}$, and $a \in \mathcal{A}_t$, the reward $R_{t,c,a}$ satisfies that $\mathbb{E}[R_{t,c,a}|\phi, \theta_t] = \phi(c, a)^\top \theta_t$.

### 3.2 Policy and Performance

Let $\mathcal{H}$ denote the set of all sequences of a finite number of action-observation pairs. Specifically, the observation at timestep 0 consists of only the initial context and available action set, and each following observation consists of a reward, a context, and an available action set. We refer to the elements of $\mathcal{H}$ as *histories*. We next introduce a policy.

**Definition 2.** A policy $\pi : \mathcal{H} \to \mathcal{P}(\mathcal{A})$ is a function that maps each history in $\mathcal{H}$ to a probability distribution over the action set $\mathcal{A}$.

A policy $\pi$ assigns, for each realization of history $h \in \mathcal{H}$, a probability $\pi(a|h)$ of choosing an action $a$ for all $a \in \mathcal{A}$. We require that $\pi(a|h) = 0$ for $a \notin \mathcal{A}_t$, where $\mathcal{A}_t$ is the available action set defined by $h$. For any policy $\pi$, we use $A_t^\pi$ to denote the action selected at time $t$ by an agent that executes policy $\pi$, and $H_t^\pi$ to denote the history generated at timestep $t$ as an agent executes policy $\pi$. Specifically, we let $H_0^\pi$ be the empty history. We let $A_t^\pi$ be such that $\mathbb{P}(A_t^\pi \in \cdot|H_t^\pi) = \pi(\cdot|H_t^\pi)$ and that $A_t^\pi$ is independent of $\{C_t\}_{t \in \mathbb{N}}$, $\{R_t\}_{t \in \mathbb{N}}$, and $\{\mathcal{A}_t\}_{t \in \mathbb{N}}$ conditioned on $H_t^\pi$, and let $H_{t+1}^\pi = (C_0, \mathcal{A}_0, A_0^\pi, R_{1,C_0,A_0^\pi}, \dots, A_t^\pi, R_{t+1,C_t,A_t^\pi}, C_{t+1}, \mathcal{A}_{t+1})$.

For all policies $\pi$, all bandits $\mathcal{E} = (\{R_t\}_{t \in \mathbb{N}}, \{C_t\}_{t \in \mathbb{N}}, \{\mathcal{A}_t\}_{t \in \mathbb{N}})$, and $T \in \mathbb{N}$, the expected cumulative reward and the long-run average expected reward are

$$\text{Return}(\mathcal{E}; T; \pi) = \sum_{t=0}^{T-1} \mathbb{E}\left[R_{t+1,C_t,A_t^\pi}\right]; \overline{\text{Return}}(\mathcal{E}; \pi) = \limsup_{T \to +\infty} \frac{1}{T} \text{Return}(\mathcal{E}; T; \pi).$$

The average expected reward is particularly useful in evaluating agent performance when both the reward process $\{R_t\}_{t \in \mathbb{N}}$ and the context process $\{C_t\}_{t \in \mathbb{N}}$ are stationary stochastic processes. In such cases, $\overline{\text{Return}}(\mathcal{E}; \pi) = \mathbb{E}\left[R_{t+1,C_t,A_t^\pi}\right]$, which is independent of $t$.

## 4 Neural Predictive Ensemble Sampling

In this section, we introduce a novel algorithm for non-stationary contextual bandit learning. The algorithm has several salient features below. See visualization of the architecture in Fig. 2

**Use Deep Neural Network Ensemble as Uncertainty Representation for Exploration.** In contextual bandit learning, an agent should intelligently balance exploration and exploitation. Thompson sampling (TS) (Thompson, 1933) stands as one of the most popular bandit learning algorithms, backed by well-established theoretical guarantees (Agrawal and Goyal, 2012; Russo and Van Roy, 2014) and good empirical performance (Chapelle and Li, 2011; Zhu and Van Roy, 2023b). To adopt TS in complex settings, Ensemble sampling (Lu and Van Roy, 2017) is introduced an efficient approximation and is also compatible with deep neural networks. Importantly, ensemble sampling has shown both theoretical effectiveness and superior empirical performance with neural networks (Lu et al., 2018; Qin et al., 2022; Osband et al., 2016). Therefore, we adopt a deep ensemble architecture.

**Seek Out Lasting Information.** In a non-stationary environment, a continuous stream of new information emerges. As an agent strives to balance between exploration and exploitation, an important consideration involves prioritizing the acquisition of information that remains relevant for a longer period of time (Liu et al., 2023). We introduce an algorithm that effectively prioritizes seeking such lasting information. Notably, our algorithm, NeuralPES, avoids the introduction of assumptions on how the rewards are generated or that of additional tuning parameters to adjust the extent of exploration. Indeed, it determines the exploration extent by training a deep neural network. To our knowledge, NeuralPES is the first algorithm that both suitably prioritizes seeking lasting information and scales to complex environments of practical interest.

### 4.1 Neural Ensemble Sampling

Before delving into the specific design of our algorithm, let us introduce a baseline algorithm which can be thought of as a deep neural network-based TS. This algorithm is referred to as the Neural Ensemble Sampling (NeuralEnsembleSampling).

At each timestep $t \in \mathbb{N}$, a NeuralEnsembleSampling agent (See Algorithm 1):

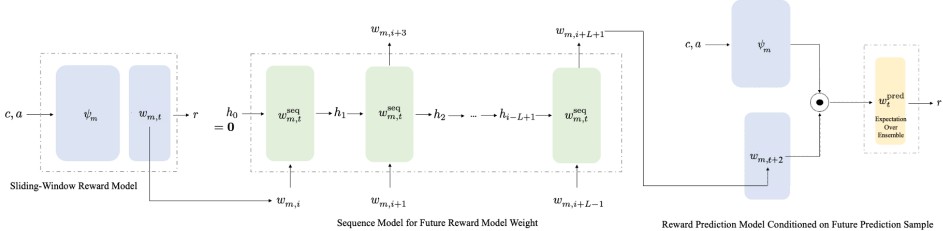

Figure 2: Visualization of three components of NeuralPES: reward, sequence, and predictive model.

1. Trains an ensemble of $M$ reward models, updating weights using stochastic gradient descent.

2. Samples $m \sim \mathrm{unif}(\{1, ..., M\})$, and uses the $m$-th reward model to predict a reward at the next timestep $\hat{R}_{t+1,C_t,a}$.

3. Selects an action that maximizes $\hat{R}_{t+1,C_t,a}$.

---

**Algorithm 1:** NeuralEnsembleSampling

---

1 **Input:** Horizon $T$, number of particles in each ensemble $M$, loss function $\mathcal{L}$, replay buffer size $K$, sequence model input size $L$, number
   of gradient steps $\tau$, $\tau_{\mathrm{seq}}$, step sizes $\alpha$, $\alpha_{\mathrm{seq}}$, minibatch sizes $K'$,
2 **Initialize:** Let replay buffer $\mathcal{B} = \emptyset$, and randomly initialize weights $\psi_{1:M}$, $w_{1:M,0}$, and $w^{\mathrm{seq}}_{1:M,0}$,
3 **for** $t = 0, 1, \ldots, T - 1$ **do**
4     **for** $m = 1, 2, \ldots, M$ **do**
5         Let $(\psi_m, w_{m,t}) \leftarrow \mathrm{TrainRewardNN}(\mathcal{B}, \mathcal{L}, \psi_m, w_{m,t-1}, \tau, \alpha, K')$
6         **sample**: $m \sim \mathrm{unif}(\{1, ..., M\})$
7         **select**: $A_t \in \arg\max_{a \in \mathcal{A}_t} f(w_{m,t}; b(\psi_m; C_t, a))$
8     **observe**: $R_{t+1,C_t,A_t}$, $C_{t+1}$, $\mathcal{A}_{t+1}$
9     **update**: Update $\mathcal{B}$ to keep the most recent $K$ tuples of context, action, reward, and timestep data.

---

| **Algorithm 2:** TrainRewardNN | **Algorithm 3:** TrainSequenceNN |
|---|---|
| 1 **Input:** Replay buffer $\mathcal{B}$, loss function $\mathcal{L}$, base network weights $\psi$, last-layer weights $w$, number of gradient steps $\tau$, step size $\alpha$, minibatch size $K'$. | 1 **Input:** Replay buffer $\mathcal{B}$, sequence model weights $w^{\mathrm{seq}}$, historical last-layer weights $w_{1:t-1}$, number of gradient steps $\tau$, step size $\alpha$, number of future steps $x$. |
| 2 **for** $i = 0, 1, \ldots, \tau - 1$ **do** | 2 **for** $i = 0, 1, \ldots, \tau - 1$ **do** |
| 3   **sample:** a minibatch $\mathcal{B}'$ of size $K'$ from replay buffer $\mathcal{B}$ | 3   **sample**: $j \sim \mathrm{unif}(\{L, ..., t - 1\})$ |
| 4   **update** $(\psi, w)$ following Equation 1 | 4   **update** $w^{\mathrm{seq}}$ following Equation 2 |
| 5 **return:** $\psi, w$ | 5 **return:** $w^{\mathrm{seq}}$ |

**The Reward Model** Figure 2 presents a visualization of the ensemble of reward models. The ensemble has $M$ particles, each consists of a base network $b$ defined by weights $\psi_m$, and last layer $f$ defined by weights $w_{m,t}$. Each particle in the ensemble is a reward model that aims to predict the reward $R_{t+1,c,a}$ given context and action pair $(c, a)$. Specifically, at each timestep $t \in \mathbb{N}$, the $m$-th reward model predicts $f(w_{m,t}; b(\psi_m; c, a))$.

We maintain a replay buffer $\mathcal{B}$ of the most recent $K$ tuples of context, action, reward, and timestep data. At each timestep, the network weights $w_{1:M}$ and $\psi_{1:M}$ are trained via repeatedly sampling a minibatch $\mathcal{B}'$ of size $K'$, and letting

$$(\psi_m, w_m) \leftarrow (\psi_m, w_m) - \alpha \sum_{(c,a,r,j) \in \mathcal{B}'} \nabla_{(\psi_m, w_m)} \mathcal{L}(f(w_m; b(\psi_m; c, a)), r) \tag{1}$$

for each $m \in [M]$. Note that we use $w_{m,t}$ to denote the last-layer weight of the $m$-th particle at the $t$-th timestep; when it is clear that we are considering a single timestep, we drop the subscript $t$.

## 4.2 PREDICTING FUTURE REWARD VIA SEQUENCE MODELING

Given the non-stationarity of the environment, a natural choice to adapt to the changing dynamics is to predict future reward model weights via sequence models, and use the predictive future reward model to select actions. We refer to this agent as the Neural Sequence Ensemble agent

At each timestep $t \in \mathbb{N}$, a Neural Sequence Ensemble agent proceeds as the following:

1. Trains an ensemble of $M$ reward models and an ensemble of $M$ sequence models through updating their weights using stochastic gradient descent.

2. Samples $m \sim \text{unif}(\{1, ..., M\})$, uses the $m$-th sequence model to predict a future reward model one step ahead of time based on past reward models, and uses this predicted future model to predict a reward at the next timestep $\hat{R}_{t+1,C_t,a}$.

3. Selects an action that maximizes $\hat{R}_{t+1,C_t,a}$.

**The Sequence Model** Figure 2 presents a visualization of the ensemble of the sequence models as well. The ensemble consists of $M$ particles. Each particle is a sequence model implemented as a recurrent neural network that aims to predict future reward model weights $w_{m,t+1}$ given historical ones $w_{m,t-L+1:t}$. At each timestep $t \in \mathbb{N}$, the $m$-th sequence model predicts $f^{\text{seq}}(w_{m,t}^{\text{seq}}; w_{m,t-L+1,t})$. The network weights $w_m^{\text{seq}}$ are trained via repeatedly sampling $j$ from $\{L, ..., t-1\}$ and letting

$$w_m^{\text{seq}} \leftarrow w_m^{\text{seq}} - \alpha \nabla_{w_m^{\text{seq}}} \mathcal{L}_{\text{MSE}}(f^{\text{seq}}(w_m^{\text{seq}}; w_{m,j-L+1:j}), w_{m,j+1}). \tag{2}$$

### 4.3 NEURAL PREDICTIVE ENSEMBLE SAMPLING

Let us now present NeuralPES. A key distinction between this algorithm and NeuralEnsemble lies in its ability to prioritize information that maintains relevance over a longer period of time. This is achieved through incorporating a new model which we refer to as the predictive model. Specifically, the predictive model is designed to take a function of a context-action pair $(c, a)$ and a future reward model as input. Its purpose is to generate a prediction for the upcoming reward $R_{t+1,c,a}$. When maintaining an ensemble of predictive models for exploration, an agent can suitably prioritize information based on how lasting the information is.

At each timestep $t \in \mathbb{N}$, a NeuralPES agent (see Algorithm 4):

1. Trains an ensemble of $M$ reward models, an ensemble of $M$ sequence models, and an ensemble of $M$ predictive models

2. Samples $m \sim \text{unif}(\{1, ..., M\})$, and uses the $m$-th sequence model to predict a future reward model two steps ahead of time based on past models.

3. Takes this predicted future model as part of input to the $m$-th predictive model, and predicts a reward at the next timestep $\hat{R}_{t+1,C_t,a}$.

4. Selects an action that maximizes $\hat{R}_{t+1,C_t,a}$.

---

**Algorithm 4:** NeuralPES

1 **Input:** Horizon $T$, number of particles in each ensemble $M$, loss function $\mathcal{L}$, replay buffer size $K$, sequence model input size $L$, number of gradient steps $\tau$, $\tau_{\text{seq}}$, $\tau_{\text{pred}}$, step sizes $\alpha$, $\alpha_{\text{seq}}$, $\alpha_{\text{pred}}$, minibatch sizes $K'$, $K''$.
2 **Initialize:** Let replay buffer $\mathcal{B} = \emptyset$, and randomly initialize weights $\psi_{1:M}$, $w_{1:M,0}$, $w_{1:M,0}^{\text{seq}}$, and $w_{1:M,0}^{\text{pred}}$
3 **for** $t = 0, 1, \ldots, T-1$ **do**
4     **for** $m = 1, 2, \ldots, M$ **do**
5         Let $(\psi_m, w_{m,t}) \leftarrow \text{TrainNN}(\mathcal{B}, \mathcal{L}, \psi_m, w_{m,t-1}, \tau, \alpha, K')$
6         Let $w_{m,t}^{\text{seq}} \leftarrow \text{TrainSequenceNN}(\mathcal{B}, w_{m,t-1}^{\text{seq}}, w_{m,1:t-1}, \tau_{\text{seq}}, \alpha_{\text{seq}}, 2)$
7         Let $w_{m,t}^{\text{pred}} \leftarrow \text{TrainPredictiveNN}(\mathcal{B}, \mathcal{L}, w_{m,t-1}^{\text{pred}}, \psi_m, w_{m,1:t-1}, \tau_{\text{pred}}, \alpha_{\text{pred}}, K'')$
8     **sample:** $m \sim \text{unif}(\{1, ..., M\})$
9     **roll out:** $\hat{w}_{m,t+2} = f^{\text{seq}}(w_{m,t}^{\text{seq}}; w_{m,t-L+1:t})$
10     **select:** $A_t \in \arg\max_{a \in \mathcal{A}_t} \sum_{i=1}^M f^{\text{pred}}(w_{i,t}^{\text{pred}}; (\hat{w}_{m,t+2} \odot b(\psi_m; C_t, a))$
11     **observe:** $R_{t+1,C_t,A_t}, C_{t+1}, \mathcal{A}_{t+1}$
12     **update:** Update $\mathcal{B}$ to keep the most recent $K$ tuples of context, action, reward, and timestep data.

---

**Algorithm 5:** TrainPredictiveNN

1 **Input:** Replay buffer $\mathcal{B}$, loss function $\mathcal{L}$, base network weights $\psi$, historical last-layer weights $w_{1:t-1}$, number of gradient steps $\tau$, step size $\alpha$, minibatch size $K''$.
2 **for** $i = 0, 1, \ldots, \tau - 1$ **do**
3     **sample:** a minibatch $\mathcal{B}''$ of size $K''$ from replay buffer $\mathcal{B}$
4     **update:** $w^{\text{pred}} \leftarrow w^{\text{pred}} - \alpha \sum_{(c,a,r,j) \in \mathcal{B}''} \nabla_{w^{\text{pred}}} \mathcal{L}(f^{\text{pred}}(w^{\text{pred}}; w_{j+2} \odot b(\psi; c, a)), r)$
5 **return:** $w^{\text{pred}}$

---

**The Predictive Model** Figure 2 also presents a visualization of the ensemble of the predictive models. The ensemble consists of $M$ particles. Each particle in the ensemble is a predictive model that aims to predict the next reward $R_{t+1,c,a}$ provided context-action pair $(c, a)$ and a future reward model of two timesteps ahead of time. Specifically, at each timestep $t \in \mathbb{N}$, the $m$-th predictive model aims to predict $R_{t+1,c,a}$ by taking an intermediate representation, i.e., $\hat{w}_{m,t+2} \odot b(\psi; c, a)$, as input.

We maintain a replay buffer $\mathcal{B}$ of the most recent $K$ tuples of context, action, reward, and timestep data. The network weights $w_{1:M}^{\text{pred}}$ are trained via repeatedly sampling a minibatch $\mathcal{B}''$ of size $K''$

$$w_m^{\text{pred}} \leftarrow w_m^{\text{pred}} - \alpha \sum_{(c,a,r,j) \in \mathcal{B}''} \nabla_{w_m^{\text{pred}}} \mathcal{L}(f^{\text{pred}}(w_m^{\text{pred}}; w_{m,j+2} \odot b(\psi_m; c, a)), r) \tag{3}$$

for each $m \in [M]$. Note that we use $w_{m,t}^{\text{pred}}$ to denote the last-layer weight of the $m$-th particle at the $t$-th timestep; when it is clear that we are considering a single timestep, we drop the subscript $t$.

**Regularization to Address Loss of Plasticity** To address the loss of plasticity, we regularize each particle's weight towards its initial weight in the last layer of the reward model ensemble and the predictive model ensemble Kumar et al. (2023). (1) and (3) now becomes

$$(\psi_m, w_m) \leftarrow (\psi_m, w_m) - \alpha \sum_{(c,a,r,j) \in \mathcal{B}'} \nabla_{(\psi_m, w_m)} \{\mathcal{L}(f(w_m; b(\psi_m; c, a)), r) + \|w_m - w_{m,0}\|_2\}.$$

$$w_m^{\text{pred}} \leftarrow w_m^{\text{pred}} - \alpha \sum_{(c,a,r,j) \in \mathcal{B}''} \nabla_{w_m^{\text{pred}}} \left\{ \mathcal{L}(f^{\text{pred}}(w_m^{\text{pred}}; w_{m,j+2} \odot b(\psi_m; c, a)), r) + \|w_m^{\text{pred}} - w_{m,0}^{\text{pred}}\|_2 \right\}. \tag{4}$$

## 4.4 THEORETICAL INSIGHTS AND ANALYSIS

We provide intuition and evidence that NeuralPES's prioritizes the acquisition of lasting information.

### 4.4.1 NEURALPES PRIORITIZES LASTING INFORMATION

We focus on comparing NeuralPES and NeuralEnsemble in linear contextual bandits. In such contexts, NeuralPES can be viewed as a neural network-based implementation of an algorithm which we refer to as linear predictive sampling (LinPS); NeuralEnsemble can be viewed as a neural network-based implementation of TS. In a linear contextual bandit, a LinPS agent carries out the following three-step procedure at each timestep, and a TS agent carries out a similar procedure, replacing $\theta_{t+2}$ with $\theta_{t+1}$:

1. samples $\hat{\theta}_{t+2}$ from the posterior $\mathbb{P}(\theta_{t+2} \in \cdot | H_t)$, and $\hat{\phi}_t$ from the posterior $\mathbb{P}(\phi \in \cdot | H_t)$.

2. estimates the reward $\hat{R}_{t+1,C_t,a} = \mathbb{E}[R_{t+1,C_t,a} | H_t, \phi = \hat{\phi}_t, \theta_{t+2} = \hat{\theta}_{t+2}]$,

3. and selects an action that maximizes the sample $A_t \in \arg\max_{a \in \mathcal{A}_t} \hat{R}_{t+1,C_t,a}$.

The procedures are carried out by approximating $\mathbb{P}(\theta_{t+1} \in \cdot | H_t)$ using the ensemble of the last layers of the reward models, approximating $\mathbb{P}(\phi \in \cdot | H_t)$ using the ensemble of the base models, approximating $\mathbb{P}(\theta_{t+2} \in \cdot | H_t)$ utilizing the sequence models; the reward estimation step of LinPS utilizes the predictive models.

To compare the behaviors of NeuralPES and NeuralEnsemble, we can compare LinPS with TS. It is worth noting that both algorithms trade off exploration and exploitation in a similar fashion, yet TS trades off between optimizing the immediate reward and learning about $\phi$ and $\theta_{t+1}$ and LinPS trades off between optimizing the immediate reward and learning about $\phi$ and $\theta_{t+2}$. If $\theta_{t+1} = \theta_t$ for all $t \in \mathbb{N}$, then the environment is stationary and the two algorithms are equivalent. In general, compared with $\theta_{t+1}$, $\theta_{t+2}$ better represents valuable information that is helpful for making future decisions. Aiming to learn about $\theta_{t+2}$, LinPS strategically prioritizes information that is still valuable in the next timestep and does not acquire information for which its value immediately vanishes.

### 4.4.2 THEORETICAL ANALYSIS

Next, we present a regret analysis that offers further evidence of LinPS's effectiveness in prioritizing lasting information. In particular, we demonstrate that LinPS excels in environments where a substantial amount of information is transient. This success stems from its strategic approach to acquire less of such information. We assume that the action set is known and remains unchanged, $\mathcal{A}_t = \mathcal{A}$ for all $t \in \mathbb{N}$, and that $\phi$ is known. We first introduce the notion of regret.

**Definition 3 (Regret).** For all policies $\pi$ and $T \in \mathbb{N}$, the regret and long-run average regret associated with a policy $\pi$ over $T$ timesteps in a linear contextual bandit is $\text{Regret}(T; \pi) = \sum_{t=0}^{T-1} \mathbb{E}\left[R_{t+1,*} - R_{t+1,C_t,A_t^\pi}\right]$, and $\overline{\text{Regret}}(\pi) = \limsup_{T \to +\infty} \frac{1}{T}\text{Regret}(T; \pi)$, respectively, where $R_{t+1,*} = \max_{a \in \mathcal{A}} \mathbb{E}[R_{t+1,C_t,a} | \theta_t]$.

We use $\text{Regret}(T)$ and $\overline{\text{Regret}}$ to denote the regret of LinPS and present a regret bound on LinPS.

**Theorem 1. (LinPS Regret Bound)** *In a linear contextual bandit, suppose $\{\theta_t\}_{t \in \mathbb{N}}$ is a reversible Markov chain. For all $T \in \mathbb{N}$, the regret and the long-run average regret of LinPS is upper-bounded by* $\text{Regret}(T) \leq \sqrt{\frac{d}{2}T\left[\mathbb{I}(\theta_2; \theta_1) + (T-1)\mathbb{I}(\theta_3; \theta_2 | \theta_1)\right]}$ *and* $\overline{\text{Regret}} \leq \sqrt{\frac{d}{2}\mathbb{I}(\theta_3; \theta_2 | \theta_1)}$.

The key proof idea essentially follows from that of (Liu et al., 2022) and (Russo and Van Roy, 2016). For the sake of completeness, we include the proof in the appendix.

It is worth noting that when $\theta_{t+1} = \theta_t$ for all $t \in \mathbb{N}_0$, we have $\text{Regret}(T) \leq \sqrt{\frac{d}{2}T\mathbb{H}(\theta_1)}$. We recover a regret bound for TS in a stationary linear contextual bandit. In the other extreme, if $\theta_t$ changes very frequently, say if $\{\theta_t\}_{t \in \mathbb{N}}$ is an i.i.d. sequence each with non-atomic distribution, then the regret of LinPS is zero that LinPS achieves optimal. This suggests that when information about $\theta_t$ is not lasting, LinPS stops acquiring this information and is optimal.

To specialize the bound to a particular example, we introduce linear contextual bandits with abrupt changes. Similar models were introduced by (Mellor and Shapiro, 2013) and (Liu et al., 2023).

**Example 2** (**Linear Contextual Bandit with Abrupt Changes**). For all $i \in [d]$, let $q_i \in [0, 1]$, and $\{B_{t,i}\}_{t \in \mathbb{N}}$ be an i.i.d. sequence of Bernoulli r.v.'s each with success probability $q_i$. For all $i \in [d]$, let $\{\beta_{t,i}\}_{t \in \mathbb{N}}$ be an i.i.d. sequence. Consider a linear contextual bandit where for all $i \in [d]$, $\theta_{1,i} = \beta_{1,i}$, and $\{\theta_{t,i}\}_{t \in \mathbb{N}}$ transitions according to $\theta_{t+1,i} = B_{t,i}\beta_{t+1,i} + (1 - B_{t,i})\theta_{t,i}$.

**Corollary 1.** (**LinPS Regret Bound in Example 2**) *For all $T \in \mathbb{N}$, the regret and long-run average regret of LinPS in a linear contextual bandit with abrupt changes is upper-bounded by*

$$\text{Regret}(T) \leq \sqrt{\frac{d}{2}T\left[\sum_{i=1}^{d}(1 - q_i)\mathbb{H}(\theta_{1,i}) + (T - 1)\sum_{i=1}^{d}\left[2\mathbb{H}(q_i) + q_i(1 - q_i)\mathbb{H}(\theta_{1,i})\right]\right]}, \text{ and}$$

$$\overline{\text{Regret}} \leq \sqrt{\frac{d}{2}\sum_{i=1}^{d}\left[2\mathbb{H}(q_i) + q_i(1 - q_i)\mathbb{H}(\theta_{1,i})\right]}, \text{ where } \mathbb{H}(q_{t,i}) \text{ denotes to the entropy of of a}$$
*Bernoulli random variable with success probability $q_{t,i}$.*

We can use Theorem 1 to investigate how the performance of LinPS depends on various key parameters of the bandit. On one hand, when $q_i = 0$ for all $i \in [d]$, i.e., when the environment is stationary, the bound becomes $\sqrt{\frac{d}{2}T\mathbb{H}(\theta_1)}$, which recovers a sublinear regret bound for TS in a stationary environment. On the other hand, as the $q_i$'s approach 1, the regret bound approaches 0, suggesting that LinPS performs well. Recall that this is a setting where $\theta_t$ are redrawn frequently, and the information associated with $\theta_t$ is not enduring. Our regret bound further confirms that LinPS continues to excel in such environments.

We consider another example, which models bandits with "smooth" changes. Similar bandits have been introduced by (Burtini et al., 2015; Gupta et al., 2011; Kuhn et al., 2015; Kuhn and Nazarathy, 2015; Liu et al., 2023; Slivkins and Upfal, 2008).

**Example 3.** [**AR(1) Linear Contextual Bandit**] **Let $\gamma \in [0, 1]^d$, with its $i$-th coordinate denoted $\gamma_i$. Consider a linear contextual bandit where $\{\theta_{t,i}\}_{t \in \mathbb{N}}$ transitions independently according to an AR(1) process with parameter $\gamma_i$: $\theta_{t+1,i} = \gamma_i\theta_{t,i} + W_{t+1,i}$, where $\{W_{t,i}\}_{t \in \mathbb{N}}$ is a sequence of i.i.d. $\mathcal{N}(0, 1 - \gamma_i^2)$ r.v.'s and $\theta_{1,i} \sim \mathcal{N}(0, 1)$.**

Applying Theorem 1 to an AR(1) linear contextual bandit, we establish the following result.

**Corollary 2.** (**LinPS Regret Bound in AR(1) Linear Contextual Bandit**) *For all $T \in \mathbb{N}$, the regret and long-term average regret of LinPS in an AR(1) linear contextual bandit is upper-bounded by* $\text{Regret}(T) \leq \sqrt{\frac{d}{4}T\left[\sum_{i=1}^{d}\log\left(\frac{1}{1 - \gamma_i^2}\right) + \sum_{t=1}^{T-1}\sum_{i=1}^{d}\log\left(1 + \gamma_i^2\right)\right]}, \overline{\text{Regret}}(T) \leq$
$\sqrt{\frac{d}{4}\sum_{i=1}^{d}\log\left(1 + \gamma_i^2\right)}$ *if $\gamma_i < 1$ for all $i \in [d]$.*

The regret bound suggests that LinPS prioritizes the acquisition of lasting information. Specifically, when $\gamma_i = 0$ for all $i \in [d]$, information about all $\theta_{t,i}$'s lose their usefulness immediately. In such contexts, LinPS achieves 0 regret and is such optimal. In addition, the regret of LinPS remains small when $\gamma_i$ is small for each $i \in [d]$, suggesting that the algorithms consistently performs well when information about $\theta_{t,i}$'s are not durable.

## 5 EXPERIMENTS

In this section, we introduce AR(1) contextual logistic bandit experiment and two experiments built on real-world data. Among the two real-world dataset experiments, one leverages one-week user interactions on Microsoft News website in time order and the other is built on Kuai's short-video platform's two-month user interaction data in time order. We consider Neural Ensemble (Osband

| Algorithm | AR(1) Average Reward | MIND 1-week Average CTR | Kuai 2-month Average Rating |
|---|---|---|---|
| Neural Ensemble | $0.5683 \pm 0.0025$ | $0.1503 \pm 0.0013$ | $1.2614 \pm 0.0017$ |
| Window Neural Ensemble | $0.5688 \pm 0.0025$ | $0.1513 \pm 0.0012$ | $1.3187 \pm 0.0023$ |
| Neural LinUCB | $0.5684 \pm 0.0020$ | $0.1468 \pm 0.0015$ | $1.2798 \pm 0.0020$ |
| Window Neural LinUCB | $0.5730 \pm 0.0031$ | $0.1482 \pm 0.0020$ | $1.3172 \pm 0.0023$ |
| Neural Linear | $0.5701 \pm 0.0027$ | $0.1467 \pm 0.0016$ | $1.2690 \pm 0.0020$ |
| Window Neural Linear | $0.5741 \pm 0.0029$ | $0.1492 \pm 0.0015$ | $1.3171 \pm 0.0026$ |
| NeuralPES | $\mathbf{0.5850 \pm 0.0023}$ | $\mathbf{0.1552 \pm 0.0013}$ | $\mathbf{1.3421 \pm 0.0016}$ |

Table 1: Empirical Experiment Results

et al., 2016), Neural LinUCB (Xu et al., 2022) and Neural Linear (Riquelme et al., 2018) and their sliding window versions (Cheung et al., 2019; 2022; Garivier and Moulines, 2008; Russac et al., 2020; Srivastava et al., 2014; Trovo et al., 2020) (to address nonstationarity in environments) as our baselines for comparison. All experiments are performed on AWS with 1 A100 40GB GPU per experiment, each with 8 CPUs, and each experiment repeated over 20 distinct seeds. To scale the experiments to the large scale experiments, we learn every batch of interactions instead of per interaction, more details in Appendix B.0.1. Constrained by computation, we do not consider Neural UCB (Zhou et al., 2020) and Neural TS (Zhang et al., 2020), given their computation requirement of inverting square matrices with dimensions equal to neural network parameter count.

## 5.1 AR(1) CONTEXTUAL LOGISTIC BANDIT

Following Example 3, An AR(1) contextual logistic bandit changes its reward function to $R_{t,c,a} \sim$ Bernoulli $\left( \sigma \left( \phi(c, a)^\top \theta_t \right) \right)$, all others the same. We set number of actions to 10, and $d = 10$, $\gamma_i = 0.99^i$. Each entry in $\theta$ is initialized with $\mathcal{N}(0, 0.01)$. Hyperparameters of the agents are presented in Appendix B.0.2. The average reward is presented in Table 1, and Figure 3a.

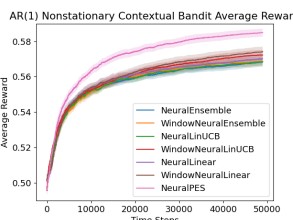

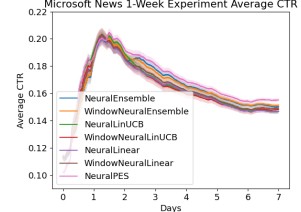

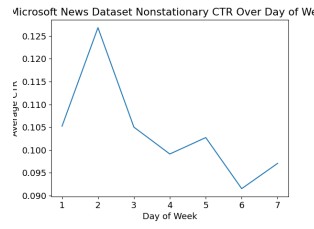

(a) AR(1) Nonstationary Contextual Bandit Average Reward

(b) Microsoft News 1-Week Experiment Average CTR

(c) Microsoft News Day of Week Nonstationary CTR

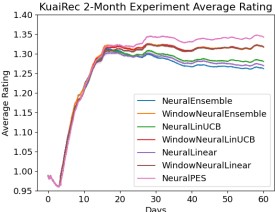

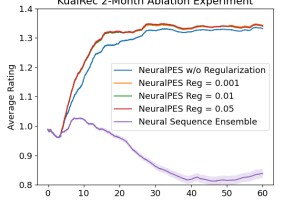

(d) KuaiRec 2-Month Experiment Average Rating

(e) KuaiRec 2-Month Ablation Experiment

Figure 3: Empirical Results and Ablations

## 5.2 MICROSOFT NEWS DATASET EXPERIMENTS

We leverage the MIND dataset (Wu et al., 2020) to carry out the first real-world dataset experiment. MIND is collected from real user interactions with Microsoft News website and its public training and validation set covers the interactions from November 9 to November 15, 2019. Each row of the MIND dataset is presented as in Table 2. In this dataset, since every recommendation's groundtruth feedback is provided at a single timestamp, no counterfactual evaluation is needed. In this experiment, we feed the rows in the order of interaction timestamp to the agent for action selection to resemble the real-world nonstationarity in user preferences. The nonstationarity presented in this dataset is commonly observed as day of week patterns in real-world recommender systems. To visualize the nonstationarity in user behavior within a week, see Figure 3c to see daily average click-through rate (CTR) in the dataset to see a week of day pattern in the dataset.

Table 2: MIND Dataset Illustration

| Impression ID | User ID | Time | User Interest History | News with Labels |
|---|---|---|---|---|
| 91 | U397059 | 11/15/2019 10:22:32 AM | N106403 N71977 N97080 | N129416-0 N26703-1 N120089-1 N53018-0 |

We sample 10,000 users from the dataset and asks candidate agents to select news recommendations sequentially according to the time order of the interactions that happened in the dataset. Hyperparameters of the agents are presented in Appendix B.0.2. Features for each recommendation is derived by average pooling over the entity embeddings of each news recommendation provided by the dataset and features for each user as average pooling over features of their clicked articles. Both user and recommendation features are of size 100. The average CTR of news recommendations offered by candidate agents over 1 week is presented in Table 1, and Figure 3b, where NeuralPES outperforms all baselines. Note that since we present interactions to users sequentially according to time order, the figure presents natural day of week seasonality from the dataset.

### 5.3 KUAIREC DATASET EXPERIMENT

While the MIND dataset offers a setup to empirically test agents' performance under day of week nonstationarity, the short duration of the dataset naturally limits the possibility of observing long-term agent behaviors under nonstationarity. In this experiment, we make slight modifications to the KuaiRec dataset Gao et al. (2022) to offer a 2-month-long real-world experiment. Every row of KuaiRec offers a user ID, the timestamp, a video ID of a recommended video, and a rating derived from the user's watch duration. The dataset also offers daily features of each user and each video candidate, of dimensions 1588 and 283 respectively. In our transformed dataset, we grouped every 12 hours of recommendation to a user into a contextual bandit format where each row contains a user ID, the 12-hour window, set of videos alongside with their corresponding ratings, sorted by the 12-hour window start time. The agent's goal is select the best recommendation to each user in each window in the order of occurrence in the real-world. Hyperparameters of the agents are presented in Appendix B.0.2. The average rating of news recommendations offered by candidate agents over 2 months is presented in Table 1 and see Figure 3d and we see NeuralPES outperforms all baselines.

### 5.4 ABLATION STUDIES

#### 5.4.1 REGULARIZATION FOR CONTINUAL LEARNING

To facilitate continual learning and avoid loss of plasticity, we leverage regularization trick introduced in Eq.4 to ensure the agent continues to learn while the environment changes. See Figure 3e. The algorithm with regularization consistently outperforms its version without regularization.

#### 5.4.2 IMPORTANCE OF PREDICTIVE MODEL

We compare NeuralPES' performance against its version without the Predictive Model, Neural Sequence Ensenble, introduced in Section 4.2. See Figure 3e. Without the Predictive Model, the agent crashes in its performance because in nonstationary environments, the environment changes are mostly unpredictable and the predictive model is responsible for determining whether a piece of information from the sequence model prediction lasts in the future.

## 6 CONCLUSION AND FUTURE WORK

There are a few lines of future work that can extend on top of this work. First of all, this work does not consider context and state evolution as a result of actions, as mentioned in Zhu and Van Roy (2023a); Xu et al. (2023); Chen et al. (2022). As these state transition kernels can also be nonstationary, it calls for future extension of this work to address nonstationarities in reinforcement learning problems. Furthermore, to enhance the quality of future reward parameter predictions, attention mechanisms (Vaswani et al., 2017) can be potentially leveraged to further improve the performance of the models.

In this paper, we introduced a novel non-stationary contextual bandit learning algorithm, NeuralPES, which is scalable with deep neural networks and is designed to seek enduring information. We theoretically demonstrated that the algorithm effectively prioritizes exploration for enduring information. Additionally, through empirical analysis on two extensive real-world datasets spanning one week and two months respectively, we illustrated that the algorithm adeptly adapts to pronounced non-stationarity and surpasses the performance of leading stationary neural contextual bandit learning algorithms, as well as their non-stationary counterparts. We aspire that the findings and the algorithm delineated in this paper will foster the adoption of NeuralPES in real-world systems.

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

# A  TECHNICAL PROOFS

## A.1  PROOF OF THEOREM 1

We first present a general regret bound that applies to any agent.

**Theorem 2. (General Regret Bound)** *In a linear contextual bandit, suppose $\{\theta_t\}_{t\in\mathbb{N}}$ is a Markov chain. For all policies $\pi$ and $T \in \mathbb{N}$, the regret is upper-bounded by* $\mathrm{Regret}(T;\pi) \leq$
$$\sqrt{\sum_{t=0}^{T-1} \Gamma_t^\pi \left[ \mathbb{I}(\theta_2;\theta_1) + \sum_{t=1}^{T-1} \mathbb{I}(\theta_{t+2};\theta_{t+1}|\theta_t) \right]}, \text{ where } \Gamma_t^\pi = \frac{\mathbb{E}\left[R_{t+1,*} - R_{t+1,C_t,A_t^\pi}\right]^2}{\mathbb{I}\left(\theta_{t+2};A_t^\pi, R_{t+1,C_t,A_t^\pi}|H_t^\pi\right)}.$$

*Proof.* For all policies $\pi$ and $T \in \mathbb{N}$,

$$\begin{aligned}
\mathrm{Regret}(T;\pi) &= \sum_{t=0}^{T-1} \mathbb{E}[R_{t+1,*} - R_{t+1,C_t,A_t^\pi}] \\
&= \sum_{t=0}^{T-1} \sqrt{\Gamma_t^\pi \sum_{t=0}^{T-1} \mathbb{I}\left(\theta_{t+2};A_t^\pi, R_{t+1,C_t,A_t^\pi}|H_t\right)} \\
&\leq \sqrt{\sum_{t=0}^{T-1} \Gamma_t^\pi \sum_{t=0}^{T-1} \mathbb{I}\left(\theta_{t+2};A_t^\pi, R_{t+1,C_t,A_t^\pi}|H_t\right)}, \quad (5)
\end{aligned}$$

where the inequality follows from Cauchy-Schwartz.

Next, observe that for all $t \in \mathbb{N}$,

$$\begin{aligned}
\sum_{t=0}^{T-1} \mathbb{I}\left(\theta_{t+2};A_t^\pi, R_{t+1,C_t,A_t^\pi}|H_t^\pi\right) &= \sum_{t=0}^{T-1} \left[\mathbb{I}\left(\theta_{t+2};\theta_{t+1}|H_t^\pi\right) - \mathbb{I}\left(\theta_{t+2};\theta_{t+1}|H_{t+1}^\pi\right)\right] \\
&= \mathbb{I}(\theta_2;\theta_1) + \sum_{t=1}^{T-1} \left[\mathbb{I}\left(\theta_{t+2};\theta_{t+1}|H_t^\pi\right) - \mathbb{I}\left(\theta_{t+1};\theta_t|H_t^\pi\right)\right] \\
&\leq \mathbb{I}(\theta_2;\theta_1) + \sum_{t=1}^{T-1} \left[\mathbb{I}\left(\theta_{t+2},\theta_t;\theta_{t+1}|H_t^\pi\right) - \mathbb{I}\left(\theta_{t+1};\theta_t|H_t^\pi\right)\right] \\
&= \mathbb{I}(\theta_2;\theta_1) + \sum_{t=1}^{T-1} \mathbb{I}\left(\theta_{t+2};\theta_{t+1}|H_t^\pi, \theta_t\right) \\
&= \mathbb{I}(\theta_2;\theta_1) + \sum_{t=1}^{T-1} \mathbb{I}\left(\theta_{t+2};\theta_{t+1}|\theta_t\right) \quad (6)
\end{aligned}$$

where the first equality follows from $\theta_{t+2} \perp H_{t+1}^\pi|\theta_{t+1}$. By (5) and (6), we complete the proof. $\square$

To apply Theorem 2 and derive a regret bound specifically for LinPS in linear contextual bandits, we establish the subsequent result, bounding $\Gamma_t^{\pi_{\mathrm{LinPS}}}$, which we will also refer to as $\Gamma_t$ for brevity.

**Lemma 1.** *In a linear contextual bandit, suppose $\{\theta_t\}_{t\in\mathbb{N}}$ is a reversible Markov chain. For all $t \in \mathbb{N}$, the information ratio associated with LinPS satisfies $\Gamma_t \leq \frac{d}{2}$, where the information ratio for any policy $\pi$ is defined as $\Gamma_t^\pi = \frac{\mathbb{E}\left[R_{t+1,*} - R_{t+1,C_t,A_t^\pi}\right]^2}{\mathbb{I}\left(\theta_{t+2};A_t^\pi, R_{t+1,C_t,A_t^\pi}|H_t^\pi\right)}.$*

*Proof.* We use $A_t$ to denote $A_t^{\pi_{\mathrm{LinPS}}}$, $H_t$ to denote $H_t^{\pi_{\mathrm{LinPS}}}$.

For all $t \in \mathbb{N}_0$, and $h = (c_0, a_0, r_1, ..., r_{t-1}, c) \in \mathcal{H}_t$, we have

$$
\mathbb{E}[R_{t+1,*} - R_{t+1,C_t,A_t}|H_t = h] = \mathbb{E}\left[\max_{a \in \mathcal{A}} \mathbb{E}[R_{t+1,c,a}|\theta_t] - R_{t+1,c,A_t}|H_t = h\right]
$$

$$
= \mathbb{E}\left[\max_{a \in \mathcal{A}} \mathbb{E}[R_{t+1,c,a}|\theta_{t+2}] - R_{t+1,c,A_t}|H_t = h\right]
$$

$$
= \mathbb{E}\left[R_{t+1,c,A_{t,c}^*} - R_{t+1,c,A_t}|H_t = h\right], \tag{7}
$$

where the second equality follows from the reversibility of $\{\theta_t\}_{t \in \mathbb{N}}$, and $A_{t,c}^*$ is defined as $A_{t,c}^* = \arg\max_{a \in \mathcal{A}} \mathbb{E}[R_{t+1,c,a}|\theta_{t+2}]$.

In addition, for all $t \in \mathbb{N}_0$, and $h = (c_0, a_0, r_1, ..., r_{t-1}, c) \in \mathcal{H}_t$, we have

$$
\mathbb{I}\left(\theta_{t+2}; A_t, R_{t+1,C_t,A_t}|H_t = h\right) = \mathbb{I}\left(\theta_{t+2}; A_t, R_{t+1,c,A_t}|H_t = h\right)
$$

$$
\geq \mathbb{I}\left(A_{t,c}^*; A_t, R_{t+1,c,A_t}|H_t = h\right), \tag{8}
$$

where the inequality follows from the data-processing inequality.

Note that $\mathbb{E}[R_{t+1,c,a}|H_t = h, \theta_{t+1}] = \mathbb{E}[R_{t+1,c,a}|\theta_{t+1}] = \phi(c,a)^\top \theta_{t+1}$. By Proposition 2 of (Russo and Van Roy, 2016), we have

$$
\mathbb{E}\left[R_{t+1,c,A_{t,c}^*} - R_{t+1,c,A_t}|H_t = h\right]^2 \leq \frac{d}{2}\mathbb{I}\left(A_{t,c}^*; A_t, R_{t+1,c,A_t}|H_t = h\right).
$$

This, together with (7) and (8), implies that

$$
\mathbb{E}[R_{t+1,*} - R_{t+1,C_t,A_t}|H_t = h]^2 \leq \frac{d}{2}\mathbb{I}\left(\theta_{t+2}; A_t, R_{t+1,C_t,A_t}|H_t = h\right). \tag{9}
$$

Therefore, for all $t \in \mathbb{N}$,

$$
\mathbb{E}\left[R_{t+1,*} - R_{t+1,C_t,A_t}\right]^2 = \mathbb{E}\left[\mathbb{E}\left[R_{t+1,*} - R_{t+1,C_t,A_t}|H_t\right]\right]^2
$$

$$
\leq \mathbb{E}\left[\mathbb{E}\left[R_{t+1,*} - R_{t+1,C_t,A_t}|H_t\right]^2\right]
$$

$$
= \frac{d}{2}\mathbb{I}\left(\theta_{t+2}; A_t, R_{t+1,C_t,A_t}|H_t\right),
$$

where the inequality follows from Jensen's inequality, and the last equality follows from (9). $\qquad \square$

Then Theorem 1 follows directly from Theorem 2 and Lemma 1.

**Theorem 1. (LinPS Regret Bound)** *In a linear contextual bandit, suppose $\{\theta_t\}_{t \in \mathbb{N}}$ is a reversible Markov chain. For all $T \in \mathbb{N}$, the regret and the long-run average regret of LinPS is upper-bounded by* $\text{Regret}(T) \leq \sqrt{\frac{d}{2}T\left[\mathbb{I}(\theta_2; \theta_1) + (T-1)\mathbb{I}(\theta_3; \theta_2|\theta_1)\right]}$ *and* $\overline{\text{Regret}} \leq \sqrt{\frac{d}{2}\mathbb{I}(\theta_3; \theta_2|\theta_1)}$.

*Proof.* For all $T \in \mathbb{N}$, the regret of LinPS is upper-bounded by

$$
\text{Regret}(T; \pi) \leq \sqrt{\sum_{t=0}^{T-1} \Gamma_t \left[\mathbb{I}(\theta_2; \theta_1) + \sum_{t=1}^{T-1}\mathbb{I}(\theta_{t+2}; \theta_{t+1}|\theta_t)\right]}
$$

$$
= \sqrt{\frac{d}{2}T\left[\mathbb{I}(\theta_2; \theta_1) + \sum_{t=1}^{T-1}\mathbb{I}(\theta_{t+2}; \theta_{t+1}|\theta_t)\right]}
$$

$$
= \sqrt{\frac{d}{2}T\left[\mathbb{I}(\theta_2; \theta_1) + (T-1)\mathbb{I}(\theta_3; \theta_2|\theta_1)\right]},
$$

where the first inequality follows from Theorem 2, the first equality follows from Lemma 1, and the last equality follows from stationarity.

$\qquad \square$

## A.2 PROOF OF COROLLARY 1

Corollary 1 follows directly from Theorem 1 and Lemma 8 of (Liu et al., 2023).

## A.3 PROOF OF COROLLARY 2

**Corollary 2. (LinPS Regret Bound in AR(1) Linear Contextual Bandit)** *For all $T \in \mathbb{N}$, the regret and long-term average regret of LinPS in an AR(1) linear contextual bandit is upper-bounded by* $\text{Regret}(T) \leq \sqrt{\frac{d}{4}T \left[ \sum_{i=1}^{d} \log \left( \frac{1}{1-\gamma_i^2} \right) + \sum_{t=1}^{T-1} \sum_{i=1}^{d} \log \left( 1 + \gamma_i^2 \right) \right]}, \overline{\text{Regret}}(T) \leq \sqrt{\frac{d}{4} \sum_{i=1}^{d} \log \left( 1 + \gamma_i^2 \right)}$ *if $\gamma_i < 1$ for all $i \in [d]$.*

*Proof.* We use $\mathbf{h}$ to denote differential entropy. If $\gamma_i < 1$ for all $i \in [d]$, then

$$
\begin{aligned}
\mathbb{I}(\theta_3; \theta_2 | \theta_1) &= \sum_{i=1}^{d} \mathbb{I}(\theta_{3,i}; \theta_{2,i} | \theta_{1,i}) \\
&= \sum_{i=1}^{d} [\mathbf{h}(\theta_{3,i} | \theta_{1,i}) - \mathbf{h}(\theta_{3,i} | \theta_{2,i}, \theta_{1,i})] \\
&= \sum_{i=1}^{d} [\mathbf{h}(\theta_{3,i} | \theta_{1,i}) - \mathbf{h}(\theta_{3,i} | \theta_{2,i})] \\
&= \sum_{i=1}^{d} \left[ \frac{1}{2} \log \left( 2\pi e (\gamma_i^2 + 1)(1 - \gamma_i^2) \right) - \frac{1}{2} \log \left( 2\pi e (1 - \gamma_i^2) \right) \right] \\
&= \sum_{i=1}^{d} \frac{1}{2} \log \left( \gamma_i^2 + 1 \right).
\end{aligned}
$$

In addition, if $\gamma_i < 1$ for all $i \in [d]$, then

$$
\mathbb{I}(\theta_2; \theta_1) = \sum_{i=1}^{d} \mathbb{I}(\theta_{2,i}; \theta_{1,i}) = \sum_{i=1}^{d} [\mathbf{h}(\theta_{2,i}) - \mathbf{h}(\theta_{2,i} | \theta_{1,i})] = \sum_{i=1}^{d} \log \left( \frac{1}{1 - \gamma_i^2} \right).
$$

Applying Theorem 1, we complete the proof. $\qquad \square$

# B IMPLEMENTATION

### B.0.1 EXTENSION TO IMPROVE SCALABILITY

Instead of generating $w_{1:M}$ ensemble every time step, $w_{1:M}$ can be generated every $K$ steps to further improve scalability of the method. In this case, the reward model $f(w_{m, \lfloor \frac{t}{K} \rfloor}; b(\psi; c, a))$ represent a posterior sample of the average reward of context-action pair $c, a$ in the current $K$-step window. The sequence model, $f^{\text{seq}}(w_{m,j}^{\text{seq}}; w_{m,j-L+1:j})$ predicts $w_{m,j+1}$. Leveraging the sequence model for two step rollouts to obtain $\hat{w}_{m,j+1}$ and $\hat{w}_{m,j+2}$, the predictive model then predicts the average reward of context-action pair $c, a$ in the current $K$-step window conditioned on future reward by computing $f^{\text{pred}}(w_{m, \lfloor \frac{t}{K} \rfloor}^{\text{pred}}; \hat{w}_{m, \lfloor \frac{t}{K} \rfloor+1} \odot b(\psi; c, a))$. The agent samples $m \sim \text{unif}(\{1, \ldots, M\})$ and takes action with

$$
A_t \in \arg\max_{a \in \mathcal{A}} f^{\text{pred}}(w_{m, \lfloor \frac{t}{K} \rfloor}^{\text{pred}}; \hat{w}_{m, \lfloor \frac{t}{K} \rfloor+1} \odot b(\psi; c, a))
$$

### B.0.2 EXPERIMENT HYPERPARAMETERS

NeuralPES's training intervals for AR(1), Microsoft News and Kuai are set to 100, 200 and 1200 respectively. All NeuralPES agents use a lookback reward parameter window of 10.

AR(1) Contextual Logistic Bandit Experiment Hyperparameters - Table 3

| Algorithm | NN Arch | Sliding Window | LR | Sequence Model | Reg Coeff | Pred Model Arch |
|---|---|---|---|---|---|---|
| Neural Ensemble | -50 - 25 - 10- | 50,000 | 0.0001 | N/A | N/A | N/A |
| Window Neural Ensemble | -50 - 25 - 10- | 10,000 | 0.0001 | N/A | N/A | N/A |
| Neural LinUCB | -50 - 25 - 10- | 50,000 | 0.0001 | N/A | N/A | N/A |
| Window Neural LinUCB | -50 - 25 - 10- | 10,000 | 0.0001 | N/A | N/A | N/A |
| Neural Linear | -50 - 25 - 10- | 50,000 | 0.0001 | N/A | N/A | N/A |
| Window Neural Linear | -50 - 25 - 10- | 10,000 | 0.0001 | N/A | N/A | N/A |
| NeuralPES | -50 - 25- | 10,000 | 0.0001 | GRU 1-layer, 25 hidden | 0.05 | -10- |

Table 3: AR(1) Hyperparameter

| Algorithm | NN Arch | Sliding Window | LR | Sequence Model | Reg Coeff | Pred Model Arch |
|---|---|---|---|---|---|---|
| Neural Ensemble | -256 - 128- | 66,000 | 0.0001 | N/A | N/A | N/A |
| Window Neural Ensemble | -256 - 128- | 20,000 | 0.0001 | N/A | N/A | N/A |
| Neural LinUCB | -256 - 128- | 66,000 | 0.0001 | N/A | N/A | N/A |
| Window Neural LinUCB | -256 - 128- | 20,000 | 0.0001 | N/A | N/A | N/A |
| Neural Linear | -256 - 128- | 66,000 | 0.0001 | N/A | N/A | N/A |
| Window Neural Linear | -256 - 128- | 20,000 | 0.0001 | N/A | N/A | N/A |
| NeuralPES | -256 - 128- | 20,000 | 0.0001 | GRU 1-layer, 128 hidden | 0.05 | -10- |

Table 4: AR(1) Hyperparameter

Microsoft News 1-Week Experiment Hyperparameters - Table 4

KuaiRed 2-Month Experiment Hyperparameters - Table 5

| Algorithm | NN Arch | Sliding Window | LR | Sequence Model | Reg Coeff | Pred Model Arch |
|---|---|---|---|---|---|---|
| Neural Ensemble | -512 - 128- | 140,000 | 0.0001 | N/A | N/A | N/A |
| Window Neural Ensemble | -512 - 128- | 20,000 | 0.0001 | N/A | N/A | N/A |
| Neural LinUCB | -512 - 128- | 140,000 | 0.0001 | N/A | N/A | N/A |
| Window Neural LinUCB | -512 - 128- | 20,000 | 0.0001 | N/A | N/A | N/A |
| Neural Linear | -512 - 128- | 140,000 | 0.0001 | N/A | N/A | N/A |
| Window Neural Linear | -512 - 128- | 20,000 | 0.0001 | N/A | N/A | N/A |
| NeuralPES | -512 - 128- | 20,000 | 0.0001 | GRU 1-layer, 128 hidden | 0.001 | -10- |

Table 5: AR(1) Hyperparameter

