# OpenReview forum: "Non-stationary Contextual Bandit Learning via Neural Predictive Ensemble Sampling"
_ICLR.cc/2024/Conference — Submitted to ICLR 2024_

### Official Review · Reviewer_jNAd · 2023-10-17

**Soundness:** 3 good
**Presentation:** 4 excellent
**Contribution:** 3 good
**Rating:** 6
**Confidence:** 2

**Summary:**

The authors propose their NeuralPES algorithm and mention that it aids recommendation systems for real-world dynamics such as seasonal preferences. They combine a  neural network architecture with their  proposed exploration strategy which they claim can more efficiently gather valuable information in evolving environments. They empirically evaluate it on real world datasets such as the Microsoft News website, where they compare it against baselines.

**Strengths:**

* Overall the paper is well written and clear to follow.
* The theoretical analysis and ablation studies are comprehensive.

**Weaknesses:**

* Nit: Some of the plots (Figure 3) are hard to read, could be plotted more clearly.
* It would be interesting to see this evaluated in a a real world dataset, distinct from the recommender system tasks to compare it's performance in a different domain, eg dynamic pricing

**Questions:**

The limitations of the work are not clearly described. What are (if any) some of the challenges with this approach?

**Details Of Ethics Concerns:**

None so far.

---

### Official Review · Reviewer_Coy3 · 2023-10-29

**Soundness:** 3 good
**Presentation:** 3 good
**Contribution:** 2 fair
**Rating:** 3
**Confidence:** 4

**Summary:**

This paper addresses the contextual nonstationary bandit problem, presenting NeuralPES, a deep neural network-based algorithm as a solution. The algorithm is a combination of ensemble sampling and future reward prediction, achieved through sequence modeling.

In essence, NeuralPES can be interpreted as a neural network implemented version of Linear Predictive Sampling (LinPS), a model that holds regret guarantees under various nonstationary environments.

The effectiveness of this approach in dealing with nonstationarity is corroborated by experiments conducted with (i) synthetic data, (ii) a real-world dataset featuring short-term nonstationarity (one week), and (iii) a real-world dataset with long-term nonstationarity (two months).

**Strengths:**

This paper, to my knowledge, is the first to address nonstationarity by considering the rate at which information disappears in the future, taking into account the applicability to real-world data.

In real-world data, there exist high-dimensional, non-linear, and diverse features (or contexts). This paper proposes a neural network-based method and architecture that allows efficient handling of these features while effectively addressing non-stationarity (via prediction). The usefulness of this approach has been verified across various applications.

As an algorithm, it enables NN-based sequence reward modeling, which predicts future rewards based on the weight sequence of past models, and ensemble sampling, which can be applied even when it is difficult to calculate the posterior distribution, to be combined.

The authors have been able to experimentally demonstrate superior performance compared to other neural-based bandit algorithms, as well as their sliding window versions.

**Weaknesses:**

I have many concerns regarding this paper, all of which I believe to be addressed by the authors.

1. The paper lacks a clear comparative analysis between PS and NeurPS. It would be beneficial for the readers if the authors could elucidate the apparent differences between the two.

2. The authors mention that PS "suffers from their scalability" and "it does not efficiently scale", however, the specifics of these issues are not clearly outlined. I would recommend providing concrete examples or explanations to support these statements.

3. It seems that PS (Liu et al. 2023) operates on an infinite reward sequence for decision-making, yet this element is not present in the current paper. It would be interesting to know how this differs or if it is similar. Either way, a discussion on this point seems necessary.

4. The theoretical analysis may appear almost identical to that of (Liu et al. 2023). Are there any differences that the authors could highlight?

5. How much execution time is required for NeuPS's learning and inference (or decision-making)? Considering the time-intensive nature of neural network training and inference, and even more so for ensemble models (depending on the value of M), it would be important to discuss the scalability implications of the execution time.

6. Given that the authors are using an "A100 40GB GPU", if the learning time is long, this could significantly increase the cost, which is a critical factor for practical applications.

7. The authors state that "the regret of LinPS is zero" and "LinPS achieves 0 regret". Is this regret referring to long-run average regret?

8. The paper mentions that the algorithm "is such optimal". It would be helpful if the authors could clarify what they mean by "optimal", as many algorithms can achieve a long-run average regret of 0.

9. How is the ensemble size M set? Is it adjusted for each experiment? Knowing the effect of its size on reward changes and computation time seems necessary for demonstrating scalability.

10. For the long-term experiments, how is the data processed? The authors mention grouping every 12 hours of recommendation into a contextual bandit format. Does this mean that the order of recommendations presented within the 12-hour period is ignored? If so, could this introduce bias?

11. Despite the major contributions of the paper appearing to lie in the experimental section, the experiments are not written in a reproducible manner. The code is not shared, and it seems that there are parameters, such as the value of M, that are not mentioned in the paper.

**Questions:**

I hope you can answer me about the comments I wrote in Weaknesses.

---

> ### Author Response · Authors · 2023-11-22
> **Thank you for your review**
>
> 1, 2, 3. In this response, we address the question from the first three questions from the reviewer. The main advantage of NeuralPES is to offer a practical neural network based non-stationary context bandit algorithm that follows the key insight from PS, which is to only explore for enduring information in the environment. The key differences are:
>
>      a) NeuralPES only requires one-step rollout into the future instead of PS which requires sampling an infinite sequence of future rewards for all actions.
>      b) NeuralPES conditions on the one-step rollout of future world parameter prediction with a sequence model instead of sampled rewards, such that we can scale to a large space of contexts and actions.
>      c) PS does not support neural networks and assumes a known world dynamics for nonstationarity, but NeuralPES doesn’t require a known world dynamics and can adapt to any type of nonstationarity.
>
> 4. The theoretical analysis differs in that (a) we now consider contextual bandits, and (b) generalizes our analysis to linear bandits.
>
> 5. Execution/inference time for NeuralPES is 2x of standard neural networks since it requires a prediction of one-step rollout of world parameters. Training-wise, it is approximately 3x to standard ensemble neural networks that carries out Thompson-sampling-like exploration per batch of data given that the model has three stages. However, in production environments, people usually need to completely retrain models to adapt to nonstationarities and hence if we can incrementally train from the current model using NeuralPES, it would actually save a lot of resources.
>
> 6. A100 40GB GPU was the only GPU available to us. From our experiment, any GPU with 2 GB is sufficient for the experiment.
>
> 7. We refer to both the cumulative regret and the long-run average regret. We will revise those sentences to make it clear.
>
> 8. In this particular scenario, the cumulative regret is zero, so the algorithm is optimal.
>
> 9. We select M = 10 according to [1] given that a 10-particle ensemble with prior functions is sufficiently enough to approximate the posterior distribution of the reward.  Increasing to 100 or even 1000 particles does not significantly improve the performance.
>
> 10. Yes the order of recommendations presented within the 12-hour is ignored but as we do not retrain our model within the 12 hours, it would not introduce any bias since we use the same NeuralPES model to address the whole batch of 12-hour data.
>
> 11. We are limited by our organization’s policy to not share code. We are in the process of obtaining permission to share open-source code. However, all hyperparameters and datasets are publicly available as listed in the paper.
>
> [1] Osband, Ian, et al. "The neural testbed: Evaluating joint predictions." Advances in Neural Information Processing Systems 35 (2022): 12554-12565.

---

### Official Review · Reviewer_Fa2p · 2023-10-29

**Soundness:** 3 good
**Presentation:** 2 fair
**Contribution:** 2 fair
**Rating:** 3
**Confidence:** 3

**Summary:**

In this paper, the authors study a non-stationary contextual bandit problem and propose an algorithm called Neural Predictive Ensemble Sampling (NeuralPES), which is scalable with neural network structure and incorporates an exploration mechanism. The authors provide theoretical results that show the strength of their method over NeuralEnsemble, a neural network based Thompson sampling algorithm. Finally, they conducted numerical experiments and tested the effectiveness of their approach on non-stationary real-world data.

**Strengths:**

- The paper is well organized, with both theoretical results and empirical evaluations on real-world data set.
- The problem of non-stationary contextual bandits remain largely unexplored and has potential real-world significance.
- The authors provide good insights for why prioritizing last information is important in a non-stationary environment.

**Weaknesses:**

- One major weakness is that I find it hard to evaluate the significance of the theoretical results presented in this paper. It appears that the authors have only compared the performance of the LinPS algorithm against TS in a non-stationary environment, which is restricted to linear contextual bandits setting and also does not really reflect the optimality/near-optimality of the algorithm.
- It is unclear how the non-stationarity of the environment is defined. Does the proposed algorithm deal with all kinds of non-stationarity? Related to the point I raised above, the theoretical results that the authors provide are only for environments with abrupt change or AR(1) type of changes. But the paper positions itself in a way that suggests the algorithm can deal with any kind of non-stationarity.
- The current results also appear to extend from similar results established under the non-contextual bandit setting. A discussion that establishes connection between the results here and those for non-contextual bandits would be helpful.
- The presentation of the algorithms also make it difficult for readers to comprehend what each component is designed for. Currently there are 5 algorithms in the paper and the relationship among these algorithms is unclear. For example, what is TrainNN used as part of Algorithm 4? What are the connections between all of the algorithms? Why are you presenting Neural Ensemble Sampling before NeuralPES and can you highlight the differences?

**Questions:**

I also have some questions related to the theoretical results established:
- Could you provide more discussions that help readers understand the regret bounds in Theorem 1 and Corollary 1? For example, I am not sure what is $\mathbb{I}(\theta_2; \theta_1)$ in the regret bound of THM 1. In corollary 1, please also elaborate on what the entropy term represents.
- I am also unsure of the statement that says when $\theta_t$ changes very quickly, “then the regret of LinPS is zero that LinPS achieves optimal”. Could you elaborate? From the regret term defined in Definition 3, it seems that the benchmark that the algorithm is compared against is the best arm at every $t$. How could the regret be zero when the changes are even more frequent?

---

> ### Author Response · Authors · 2023-11-22
> **Thank you for your review**
>
> Thank you for the detailed feedback! We would like to make the following clarifications.
>
> Responding to Weaknesses:
>
> You raised the point that our analysis is “restricted to linear contextual bandits,” which is indeed a fairly large class of problems, and can be a good starting point.
>
> Our algorithm does not rely on any assumptions of how the rewards are generated. Our theoretical results are introduced with the purpose to provide more intuition on how our algorithm works. Our result (Corollary 1) is general, and we establish a result in AR(1) bandit (Corollary 2) only to provide more intuition on this particular example. We will add a discussion/note on theoretical results.
>
> Thank you for your suggestions on improving the presentation of the paper. Algorithm 2 is a submodule of Algorithm 1 and Algorithm 1 makes a call to Algorithm 2. Similarly, Algorithms 3 and 5 are submodules of Algorithm 2.
>
> Responding to Questions:
> Thank you for bringing this up. We use $\mathbb{I}$ to denote mutual information, and $\mathbb{I}(\theta_2; \theta_1)$ corresponds to the mutual information between $\theta_2$ and $\theta_1$. In Corollary 1, we use $\mathbb{H}(p)$ to denote the entropy of a $\mathrm{Bernoulli}(p)$ random variable. We will include definition of mutual information and entropy in our manuscript.
> The regret is always non-negative, so when the regret is zero, it implies that the algorithm is optimal. We would like to clarify that the benchmark that the algorithm is compared against is not “the best arm”, i.e., $\arg\max_{a \in \mathcal{A}}\mathbb{E}[R_{t+1, a} | \theta_{t+1}]$, but $\arg\max_{a \in \mathcal{A}}\mathbb{E}[R_{t+1, a} | \theta_t]$ instead.
>
> Therefore, when $\{\theta_t\}$ for ${t \in \mathbb{N}}$ is i.i.d. (an example when changes are frequent),  $\mathbb{E}[R_{t+1, a} | \theta_t] = \mathbb{E}[R_{t+1, a}]$ and it makes sense that regret can be zero.

---

### Official Review · Reviewer_EExp · 2023-10-30

**Soundness:** 2 fair
**Presentation:** 2 fair
**Contribution:** 2 fair
**Rating:** 3
**Confidence:** 4

**Summary:**

This paper studies the contextual bandits due to non-stationarity caused by factors like seasonality and evolving social trends. Existing algorithms either overly explore or cannot handle high-dimensional user-specific features and large action sets. The paper introduces a non-stationary contextual bandit algorithm that combines a scalable deep neural network architecture with a strategic exploration mechanism that prioritizes valuable information in a changing environment. Empirical evaluations on real-world recommendation datasets with non-stationarity show that this approach outperforms state-of-the-art baselines.

**Strengths:**

It is very interesting and necessary to extend neural bandits to the non-stationary environment. It is indeed my first time to see the neural bandit's work extending to non-stationary. The introduced algorithm is embedded with slide window to overcome the changing reward mapping.

**Weaknesses:**

However, (1) I am not very convinced by the exploration effectiveness of ensemble networks. The exploration comes from randomly draw ing neural models, but it doesn't consider the estimation confidence interval of a single neural model like UCB or TS. It looks like an ensemble of greedy models.

(2) The training cost is too huge for this approach. In linear bandits, the training of linear models can be trained quickly. But for neural models, it cannot work. For one neural model, the training cost is already huge, but the algorithm needs to train a set of neural models in each round. Especially, it is for online learning scenarios. I don't think this algorithm can be scaled to large systems in practice.

(3) The analysis is for the linear model with linear reward function, which avoids the challenge of analyzing neural networks.

**Questions:**

See weakness.

---

> ### Author Response · Authors · 2023-11-22
> **Thank you for your review.**
>
> Thank you for acknowledging that this is the first time you see the neural bandit’s work in non-stationary environments. We would like to make the following clarifications.
>
> (1) An Ensemble of neural networks can perform Thompson sampling-like exploration. This is also discussed in literature Lu and Van Roy (2017) [1], Qin et al. (2022) [2] and Osband et al. (2016) [3].
>
> (2) Neural contextual bandit algorithms are very commonly adopted in research that studies scalability [4, 5] and real-world applications [6], and also shown to be very incremental in terms of both training and inference [7]. Since each particle neural network is independent from others in an ensemble, in production systems, ensemble training usually leverages distributed GPU training and hence in general on par with standard deep supervised learning approaches. Since each ensemble particle is identical to a deep supervised learning model, inference cost is identical.
>
> (3) The purpose of our theoretical analysis is to provide supporting evidence on the efficacy of our method, and we believe it achieves its purpose. We leave the analysis involving neural networks to future work.
>
> [1] Lu, Xiuyuan, and Benjamin Van Roy. "Ensemble sampling." Advances in neural information processing systems 30 (2017).
>
> [2] Qin, Chao, et al. "An analysis of ensemble sampling." Advances in Neural Information Processing Systems 35 (2022): 21602-21614.
>
> [3] ​​Osband, Ian, et al. "Deep exploration via bootstrapped DQN." Advances in neural information processing systems 29 (2016).
>
> [4] Xu, Pan, et al. "Neural Contextual Bandits with Deep Representation and Shallow Exploration." International Conference on Learning Representations. 2021.
>
> [5] Riquelme, Carlos, George Tucker, and Jasper Snoek. "Deep bayesian bandits showdown." International conference on learning representations. Vol. 9. 2018.
>
> [6] Lu, Xiuyuan, Zheng Wen, and Branislav Kveton. "Efficient online recommendation via low-rank ensemble sampling." Proceedings of the 12th ACM Conference on Recommender Systems. 2018.
>
> [7] Zhu, Zheqing, and Benjamin Van Roy. "Scalable Neural Contextual Bandit for Recommender Systems." ACM International Conference on Information and Knowledge Management. 2023.

---

### Meta-Review · Area_Chair_5AT3 · 2023-12-12

**Metareview:**

The reviewers are overall not positive about the contribution of the work and several weaknesses of the paper has been pointed which weakens the competency of the submission. Some of the reviewers' questions are also not addressed in the rebuttal.

We urge the authors to address the reviewers' concerns before resubmitting the paper to the next suitable venue.

**Justification For Why Not Higher Score:**

N/A

**Justification For Why Not Lower Score:**

N/A

---

### Decision · Program_Chairs · 2024-01-16

Reject